# Hierarchical control of enzymatic actuators using DNA-based switchable memories

Lenny H.H. Meijer[1,2,3], Alex Joesaar[1,2,3], Erik Steur[2,4], Wouter Engelen[1,2], Rutger A. van Santen[2], Maarten Merkx[1,2] & Tom F.A. de Greef[1,2,3]

Inspired by signaling networks in living cells, DNA-based programming aims for the engineering of biochemical networks capable of advanced regulatory and computational functions under controlled cell-free conditions. While regulatory circuits in cells control downstream processes through hierarchical layers of signal processing, coupling of enzymatically driven DNA-based networks to downstream processes has rarely been reported. Here, we expand the scope of molecular programming by engineering hierarchical control of enzymatic actuators using feedback-controlled DNA-circuits capable of advanced regulatory dynamics. We developed a translator module that converts signaling molecules from the upstream network to unique DNA strands driving downstream actuators with minimal retroactivity and support these findings with a detailed computational analysis. We show our modular approach by coupling of a previously engineered switchable memories circuit to downstream actuators based on $\beta$-lactamase and luciferase. To the best of our knowledge, our work demonstrates one of the most advanced DNA-based circuits regarding complexity and versatility.

[1] Laboratory of Chemical Biology, Department of Biomedical Engineering, Eindhoven University of Technology, 5612 AZ Eindhoven, The Netherlands. [2] Institute for Complex Molecular Systems, Eindhoven University of Technology, 5612 AZ Eindhoven, The Netherlands. [3] Computational Biology Group, Department of Biomedical Engineering, Eindhoven University of Technology, 5612 AZ Eindhoven, The Netherlands. [4] Dynamics and Control Group, Department of Mechanical Engineering, Eindhoven University of Technology, 5612 AZ Eindhoven, The Netherlands. Correspondence and requests for materials should be addressed to T.F.Ad.G. (email: t.f.a.d.greef@tue.nl)

DNA has proven to be a versatile building block for the construction of functional devices useful in diagnostics and therapeutics[1], including nanostructures for the delivery of cargo[2–4], molecular walkers[5], or actuators that mechanically control protein activity[6–11]. Additionally, synthetic molecular platforms based on enzyme-free DNA strand exchange are highly amenable for the rational design of reaction networks due to the predictable thermodynamics of DNA binding, enabling the engineering of networks with functionalities such as amplification[12, 13], thresholding[14, 15], or Boolean and arithmetic operations[16–18]. Enzymatically driven DNA-based networks exhibit greater nonlinear kinetics, higher turnover rates, and thereby further increase the range of dynamic behaviors[19]. Recent work has shown that transcriptional circuits in which genelets, i.e., DNA templates that produce RNA regulators for other genelets, can yield switches[20], oscillators[21, 22], and adaptive dynamics[23]. In addition, networks based on DNA replication, nicking, and degradation have shown to be highly modular and have been engineered to display stable oscillations[24], multi-stability[25], traveling waves[26, 27], and chaotic dynamics[28]. These cell-free circuits provide a simple and well-controlled platform to implement various types of regulatory functions, which increases our understanding of the design principles underlying specific cellular tasks[29]. Interestingly, regulatory circuits with specific topology-function correlation inside living cells are not isolated but interconnected to downstream processes resulting in hierarchical layers of signal generation and processing[29]. However, coupling of enzymatically driven DNA-based networks displaying higher-order dynamics to downstream processes has rarely been reported. Franco and co-workers realized the control of a DNA tweezer using a genelet-based oscillator and demonstrated an insulating device to reduce retroactivity[22, 30]. However, to the best of our knowledge the control of enzymatic actuators by dissipative, enzymatically driven DNA circuits has not been reported. Here, we engineer and implement hierarchical control of biochemical actuators, such as a NanoLuc-based actuator[31] and a self-inhibitory TEM1 β-lactamase construct[9], using an upstream polymerase–exonuclease–nickase (PEN)-based switchable memories circuit[25]. We developed a translator module enabling the translation of the dynamic state of the upstream network to the directed control of the downstream enzymatic actuators (Fig. 1a) with minimal retroactivity[32, 33]. Our design strategy for the translator module harnesses several design criteria resulting in minimal retroactivity as validated by experiments and corroborated by a theoretical analysis. The translator module improves the utility of feedback-controlled DNA circuits, as it interfaces complex information processing molecular programs to functional downstream enzymatic processes in a modular and orthogonal fashion. By precise and careful tuning of many different enzymatic reactions and a fundamental understanding on the origin of retroactivity, we are able to demonstrate hierarchical control of enzymatic actuators by dissipative DNA circuits.

## Results

**Coupling PEN-based networks to a translator module.** Rondelez and co-workers introduced a methodology in which enzymatically enriched DNA-based networks of arbitrary complexity can be engineered in an artificial, non-living, and well-controlled setting[24]. The methodology, shown in Fig. 1b, includes activation, inhibition, and destruction of short primers (~11 nucleotides) and single-stranded DNA (ssDNA) inhibitors (~16 nucleotides) carried out by polymerase, exonuclease and nickase. Synthetic DNA templates (~22 nucleotides) are activated by input primers acting as regulatory signals for the production of ssDNA outputs. Inhibition of the activation module is achieved by binding of an inhibitor strand to the target template, preventing the input primer from binding to the respective template. Importantly, input, output, and inhibitor strands are degraded over time by exonuclease, while the template strands are protected from degradation by phosphorothioate modifications at their 5′ ends.

PEN-based networks are modular as templates can be connected so that they control each other's activity, feature out-of-equilibrium behavior, and exhibit nonlinear dynamics. These properties are essential for the construction of network topologies with higher-order dynamics, as shown in previous work[24–28]. However, the toolbox makes use of relatively short

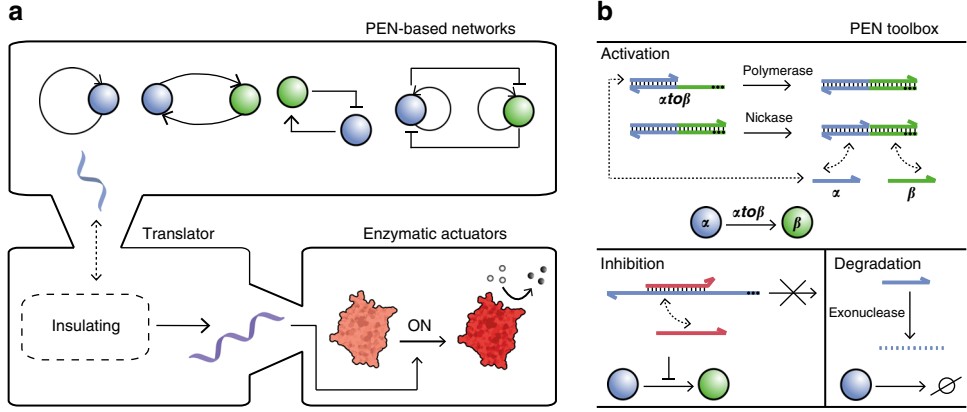

**Fig. 1** Controlling enzymatic actuators using dissipative DNA-based circuits. **a** The dynamics of PEN-based circuits is used to time and control downstream processes. In this work a translator module was designed that translates output strands from the upstream network to unique regulatory DNA strands, which in turn drive enzymatic actuators, while insulating the upstream network from the additional load. **b** Enzyme-driven DNA-based circuits based on the PEN toolbox comprise three modules, including activation, inhibition, and degradation. Activation is achieved by binding of input ssDNA signals (e.g., primer α) to their target template (e.g., *αtoβ*), which enables DNA polymerase to extend the oligomer-template pair, followed by nicking of the elongated strand. This results in the return of the input ssDNA signal and a newly formed output ssDNA (e.g., oligomer *β*), which dissociate from the template because these reactions are performed around the melting temperature of the partial duplexes. The activation of templates can be inhibited by ssDNA strands that are complementary to part of the template's sequence, and possess a two-base mismatch at their 3′-ends which prevents extension of the partial duplex, rendering the template strand inactive. Finally, signal and inhibition strands are degraded over time by exonuclease. The template strands are protected from degradation by 5′-end phosphorothioate backbone modifications indicated by the black dots

single-stranded primers that have a melting temperature around the experimental temperature of 42 °C, limiting the PEN toolbox from activation of DNA-based enzymatic actuators that typically require much longer activator strands[6–11, 31, 34]. We developed a PEN-based translator module that translates the short primers from the PEN toolbox to relatively long output DNA strands (>30 bases). Ideally, the translator module should completely isolate the upstream network from the enzymatic actuators as this would allow modular connection of PEN-based circuits to downstream processes. Inevitably, the translator module provides a load to the upstream circuit. The interconnection should therefore be designed to have a minimal effect on the dynamics of the core network, i.e., retroactivity should be minimized. Previous studies have shown that the retroactivity from a downstream system can be attenuated either by connecting the load via a large gain and/or by separation of timescales[22, 30, 32, 33, 35], i.e., the dynamics of the interface connecting the load to the upstream network should be fast compared to the intrinsic dynamics of the core network itself.

Based on these considerations the PEN-based translator module was designed to provide a high gain while only transiently sequestering the output of the upstream DNA system (Fig. 2a). Primer $\alpha$ from the upstream network reversibly binds to the 3′-end of template $\alpha toX$, with forward and backward rates (minutes, Supplementary Table 2) that are substantially faster than the timescale of the dynamics of the PEN toolbox (hours). Template $\alpha toX$ is protected from degradation by phosphorothioate modifications at its 5′-end, and hence the load to the upstream circuit is time invariant. Similar to the activation module of the

PEN toolbox, the polymerase extends $\alpha$ followed by the action of nickase resulting in a nicked duplex regenerating $\alpha$ and producing output strand $X$. While $\alpha$ reversibly dissociates from the template, $X$ is tightly bound and can only be released via DNA polymerase-mediated strand-displacement during extension of $\alpha$, which now can activate a downstream enzymatic actuator. Subsequently, the nickase hydrolyzes the upper strand of the duplex after which a new cycle starts resulting in linear amplification of $X$. Besides minimizing the retroactivity to the dynamics of the upstream reaction network, these features result in a translator module that responds fast, thereby transducing the state of the upstream network almost instantaneously (vide infra).

To provide proof-of-principle for the translator module, we characterized the performance of the translator module isolated from the upstream network (Fig. 2b–d). To this end, an experiment was performed for a concentration range of $\alpha toX$ in presence of polymerase and nickase and the output $X$ was quantified using a molecular beacon (MB). As expected, addition of $\alpha$ results in linear amplification of $X$ (Fig. 2b, Supplementary Fig. 1) eventually opening all available MBs. To quantify the kinetics and gain of the translator module in more detail, the production rate of $X$ was determined for a concentration range of $\alpha toX$ (Fig. 2b). For low concentrations of $\alpha toX$, the data shows a linear increase of the production rate, while for higher concentrations the production rate levels off. While the concentration of $\alpha$-$\alpha toX$ increases linear with $\alpha toX$ for the concentration range used in these experiments (Supplementary Table 2), saturation of the enzymes limits the rate at which strand $X$ can be produced.

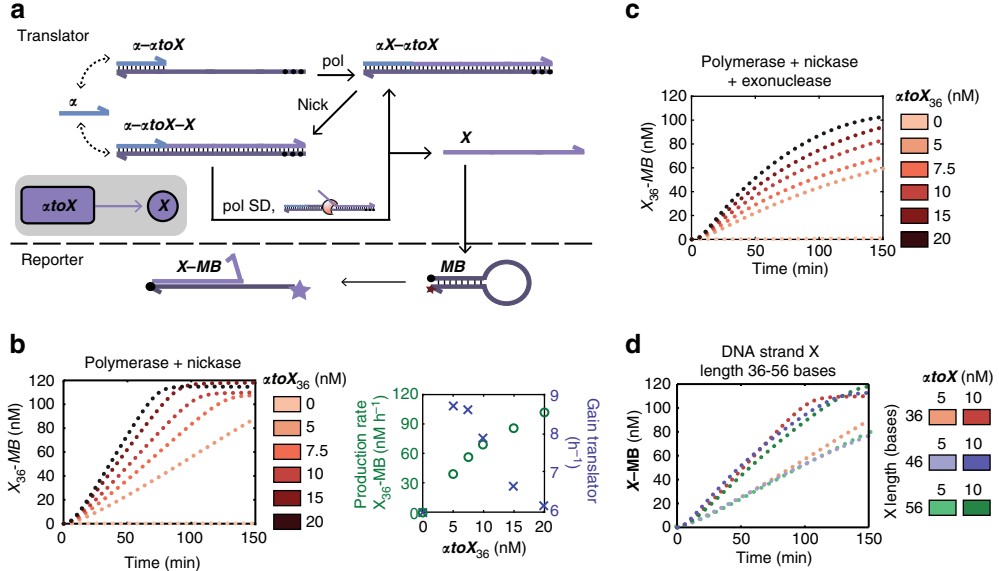

**Fig. 2** Characterization of the translator module. **a** The design of the translator module in which an output ssDNA of the upstream network (e.g., primer $\alpha$) hybridizes to the template strand ($\alpha toX$) of the translator module. After the action of polymerase (pol) and nickase (nick) the input primer reversibly dissociates, while the relatively long output strand (ssDNA $X$) is released via polymerase-mediated strand-displacement (pol SD). The reaction cycle continues resulting in linear amplification. The black dots at the 5′-end of DNA strands represent phosphorothioate backbone modifications. A simplified illustration of the translator module is shown on the left. The production of output $X$ was quantified using a molecular beacon ($MB_x$). **b** Experimental traces of the linear amplification of $X_{36}$ (36 bases) performed for a concentration range of the translator template $\alpha toX$ in presence of polymerase (15 U mL$^{-1}$) and nickase (10 U mL$^{-1}$) and initiated by addition of $\alpha$. The production rate of $X_{36}$-$MB$ and gain of the translator for the concentration range of translator template were determined from the slope of the experimental traces in the linear regime. The gain (Eq. (1)) is defined by the number of output $X$ produced per unit time (1 h) per complex of primer $\alpha$ bound to template $\alpha toX$ calculated using the thermodynamic dissociation constant (Supplementary Table 2). **c** Experimental traces of the linear amplification of $X_{36}$ (36 bases) performed for a concentration range of the translator template $\alpha toX$ in presence of polymerase (15 U mL$^{-1}$), nickase (10 U mL$^{-1}$), and exonuclease (200 nM) and initiated by addition of $\alpha$ protected with phosphorothioate modifications at its 5′-end (Supplementary Fig. 2). **d** The performance of the translator module for varying sequences and lengths of $X$. The experiment was performed starting with 10 nM of $\alpha toX$ in the presence of 15 U mL$^{-1}$ polymerase and 10 U mL$^{-1}$ nickase and initiated by addition of $\alpha$. Experiments were carried out as described in the Methods. Fluorescence was converted to concentration using a standard curve (Supplementary Fig. 17)

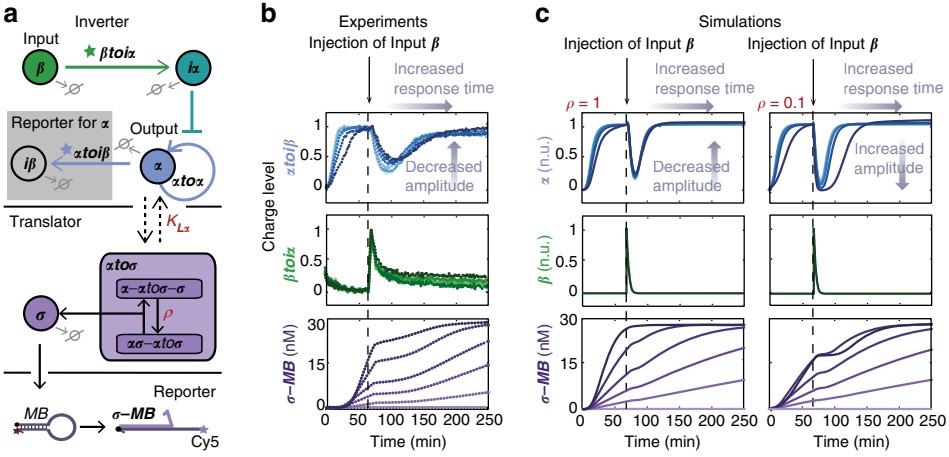

**Fig. 3** Coupling of the translator module to an upstream INVERTER network. **a** Schematic illustration of the translator module coupled to a PEN-based INVERTER network. Multiplex monitoring of the dynamics of the network is performed using endogenous template *βtoiα* and an exogenous template *αtoiβ* which are 3′-end fluorescently labeled with DY530 and FAM, respectively, while the output strand *σ* of the translator module is measured via a MB bearing a fluorophore-quencher pair. **b** Results of the experiments that were conducted for 0, 2, 5, 10, 20, and 40 nM (light to dark) of translator template *αtoσ* in the presence of 7 nM *αtoα*, 20 nM of *βtoiα* and *αtoiβ*, 30 nM MB, 10 U mL$^{-1}$ Bst 2.0 WarmStart DNA polymerase, 25 U mL$^{-1}$ Nt. bstNBI, and 50 nM ttRecJ. The INVERTER is activated by addition of 0.5 nM *α* which is initially amplified until it reaches steady-state in which production by polymerase and nickase and degradation due to exonuclease are balanced. Applying a pulse of 30 nM of input *β* at this point initiates the production of *iα*, which inhibits autocatalytic production of output *α*. As input strand *β* gets degraded the system returns its pre-stimulus steady-state. Hence, the INVERTER network shows a pulse response after injection of input *β*, which can be characterized by its amplitude and response time which is the time needed to recover to the pre-stimulus steady-state. The charge level is the normalized fluorescence of the signal of DY530 and FAM fluorophores, which is 0 in the absence of template's input primer and 1 at the maximal or steady-state value of primer *β* and *α*, respectively. The fluorescence of Cy5 fluorophore was converted to concentration of DNA strand *σ* using a standard curve (Supplementary Fig. 17). **c** Results of simulations using the heuristic model with the same concentrations of translator template as used during the experiments in **b** and for different values of *ρ*. The traces were converted to normalized units (n.u.) by normalizing *α* to the steady-state concentration and normalizing *β* to its maximum value

Next, we quantified the gain of the translator module using Eq. (1):

$$\text{Gain} = \frac{d[X]/dt}{[\alpha - \alpha toX]} \qquad (1)$$

with d[X]/dt the production rate of X in nM h$^{-1}$ and [α−αtoX] the concentration (in nM) of the partial duplex consisting of input **α** bound to template **αtoX** calculated using the thermo-dynamic dissociation constant (Supplementary Table 2). The results reveal a decreasing gain from 9 h$^{-1}$ to 6 h$^{-1}$ with increasing concentration of translator template, i.e., per hour one input produces 9–6 outputs depending on the concentration of translator template. The decrease in gain with increasing concentration of translator template is the result of the hyperbolic dependence of the production rate on the concentration of translator template. Importantly, the PEN toolbox includes an exonuclease, which degrades produced DNA strands in the reaction network. In order to test the compatibility of exonuclease with the translator module, experiments were performed in the presence of polymerase, nickase, and exonuclease (Fig. 2c). The results show that the translator module is able to amplify X even in the presence of exonuclease. Further experiments reveal that the translator module is able to produce sequences of different lengths with very similar kinetics (Fig. 2d), showing the modular performance of the translator. Based on these results we conclude that the translator module should be generally applicable to allow control of downstream DNA-templated biochemical reactions by PEN-based networks.

**Retroactivity of the translator module**. In order to assess the retroactivity that arises from connecting the translator module to

an upstream network, we first coupled the translator module to a PEN-based INVERTER circuit (Fig. 3a), consisting of DNA templates that comprise part of the bistable switch (vide infra). The INVERTER network is based on an autocatalytic module producing activator **α**, which is inhibited by addition of input **β** via inhibitor species **iα** resulting in an output that is inverted compared to the change in input. As input strand **β** gets degraded **iα** levels decrease, resulting in an increase of activity of the autocatalytic module signaling the return of the system to its pre-stimulus steady-state. Thus, the INVERTER network shows a pulse response after injection of input **β**, characterized by its amplitude and response time (Fig. 3). The dynamics of the INVERTER network were followed by N-quenching[36], which monitors oligomer hybridization to templates by a change in fluorescence. Experiments were performed with increasing concentrations of translator template **αtoσ**, while the concentration of output **σ** was assessed using a MB. As can be observed from Fig. 3b, the experimental results show increasing production of **σ** for higher concentrations of **αtoσ**. More importantly, the production of **σ** ceases upon injection of **β** and continues when the INVERTER returns to pre-stimulus steady-state, showing that the production rate of **σ** follows the dynamics of the INVERTER circuit instantaneously. In addition, the results show a very gradual change in dynamics of the INVERTER for increasing concentrations of translator template, indicating low retroactivity.

To further quantify the retroactivity that arises from coupling of the translator module to the PEN-based circuit, we constructed a minimal model that allows us to rationalize the effect of increasing loads on the dynamics of the INVERTER network. The model consists of a set of ordinary differential equations expressing the trajectories of **β**, **iα**, **α**, **σ**, **MB**, **σ-MB**, and **iβ**

(detailed description provided in Supplementary Notes):

$$[\dot{\beta}] = -\frac{V_{exo}[\boldsymbol{\beta}]}{K_{exo} + [\boldsymbol{\alpha}] + [\boldsymbol{\beta}] + [i\alpha] + [i\beta] + [\boldsymbol{\sigma}]}; \quad (2a)$$

$$[i\dot{\alpha}] = \frac{V_{ia}[\boldsymbol{\beta}]}{K_{\beta} + [\boldsymbol{\beta}]} - \frac{V_{exo}[i\alpha]}{K_{exo} + [\boldsymbol{\alpha}] + [\boldsymbol{\beta}] + [i\alpha] + [i\beta] + [\boldsymbol{\sigma}]}; \quad (2b)$$

$$[\dot{\alpha}] = \frac{V_{\alpha}[\boldsymbol{\alpha}]}{K_{\alpha} + [\boldsymbol{\alpha}] + \lambda_{\alpha}[i\alpha]} - \frac{V_{exo}[\boldsymbol{\alpha}]}{K_{exo} + [\boldsymbol{\alpha}] + [\boldsymbol{\beta}] + [i\alpha] + [i\beta] + [\boldsymbol{\sigma}]}$$
$$- \frac{V_{L\alpha}[\boldsymbol{\alpha}]}{K_{L\alpha} + [\boldsymbol{\alpha}]} + k_{\alpha}[\boldsymbol{\theta}]; \quad (2c)$$

$$[\dot{\sigma}] = \frac{V_{L\alpha}[\boldsymbol{\alpha}]}{K_{L\alpha} + [\boldsymbol{\alpha}]} - \frac{V_{exo}[\boldsymbol{\sigma}]}{K_{exo} + [\boldsymbol{\alpha}] + [\boldsymbol{\beta}] + [i\alpha] + [i\beta] + [\boldsymbol{\sigma}]} - k_{rep}[\boldsymbol{\sigma}][MB]; \quad (2d)$$

$$[\dot{MB}] = -k_{rep}[\boldsymbol{\sigma}][MB]; \qquad [\boldsymbol{\sigma} - MB] = k_{rep}[\boldsymbol{\sigma}][MB]; \quad (2e/f)$$

$$[i\dot{\beta}] = \frac{V_{i\beta}[\boldsymbol{\alpha}]}{K_{\alpha} + [\boldsymbol{\alpha}]} - \frac{V_{exo}[i\beta]}{K_{exo} + [\boldsymbol{\alpha}] + [\boldsymbol{\beta}] + [i\alpha] + [i\beta] + [\boldsymbol{\sigma}]}; \quad (2g)$$

$$[\boldsymbol{\theta}] = \rho \frac{V_{L\alpha}[\boldsymbol{\alpha}]}{K_{L\alpha} + [\boldsymbol{\alpha}]} - k_{\alpha}[\boldsymbol{\theta}]. \quad (2h)$$

The production of oligomers $\boldsymbol{\alpha}$, $i\boldsymbol{\alpha}$, $\boldsymbol{\sigma}$, and $i\boldsymbol{\beta}$ by polymerase and nickase is described by a single Michaelis–Menten approximation with the maximum rates ($V_i$) roughly controlled by the concentration of the template that encodes the corresponding oligomer. By assuming that the catalytic rate constant of the two enzymatically driven reactions is relatively small, the Michaelis–Menten parameter ($K_i$) can be approximated by the equilibrium dissociation constant of the input primer to its template. Inhibition of $\boldsymbol{\alpha}$ production depends on the concentration of the inhibitor $i\boldsymbol{\alpha}$ and the ratio of the equilibrium dissociation constants of $\boldsymbol{\alpha}$ and $i\boldsymbol{\alpha}$, denoted by $\lambda_{\alpha}$ (Eq. (2c)). Binding of $\boldsymbol{\sigma}$ to the reporter ($MB$) is described as a single step with second-order rate constant ($k_{rep}$, Eq. (2d–f)). Furthermore, degradation of input $\boldsymbol{\beta}$, output $\boldsymbol{\alpha}$, inhibitor $i\boldsymbol{\alpha}$, $i\boldsymbol{\beta}$, and $\boldsymbol{\sigma}$ is modeled by a Michaelis–Menten approximation, which includes terms that describe competition between the substrates. The two final terms in Eq. (2c) take into account the change in the concentration of $\boldsymbol{\alpha}$ caused by the reversible sequestration of $\boldsymbol{\alpha}$ by the translator module. While the second last term represents the amount of $\boldsymbol{\alpha}$ that is sequestered by the translator for production of $\boldsymbol{\sigma}$, the last term accounts for the reproduction of $\boldsymbol{\alpha}$ due to the dissociation of $\boldsymbol{\alpha}$ from the nicked state of the translator module. The rate of reproduction of $\boldsymbol{\alpha}$ is given by $\boldsymbol{\theta}$, representing the concentration of nicked translator module, linearly scaled with the dissociation rate constant of $\boldsymbol{\alpha}$ ($k_{\alpha}$). Specifically, the dissociation rate constant of $\boldsymbol{\alpha}$ is determined by the equilibrium dissociation constant ($K_{L\alpha}$) as the association rate constant is invariable for primers with lengths exceeding five bases[37]. We introduced a constant $\rho$, which models the fraction of translator module being in the nicked state ($\alpha$—$\alpha t o \sigma$—$\sigma$), depending on the timescale of nicking the duplex ($\alpha \sigma$—$\alpha t o \sigma$) relative to the timescale of the polymerase strand-displacement reaction. In the extreme case of $\rho = 1$, the equilibrium of the two states of the

translator module is shifted to the nicked state (minimal inherent retroactivity), and hence the amount of $\boldsymbol{\alpha}$ reproduced depends on $K_{L\alpha}$. In the other extreme case, i.e., $\rho = 0$, the equilibrium of the two states of the translator module is fully shifted to the duplex conformation ($\alpha \sigma$—$\alpha t o \sigma$), and therefore no $\boldsymbol{\alpha}$ is reproduced independent on $K_{L\alpha}$ (maximal inherent retroactivity). In summary, retroactivity is determined by the translator concentration, $K_{L\alpha}$ and $\rho$ which is an inherent property of the translator module. While in principle retroactivity could also arise due to global coupling arising from competition of primers for exonuclease, the influence of this effect was found to be negligible (Supplementary Fig. 3).

While the kinetic parameters and the equilibrium dissociation constants were measured in separate experiments (Supplementary Figs. 18–21 and Supplementary Table 2) the parameter $\rho$ is defined by the system-dependent enzyme competition between polymerase and nickase (Supplementary Notes and Supplementary Fig. 30). Figure 3c displays the dynamics of the INVERTER and translator using the heuristic model for two values of $\rho$. The simulations show that coupling of the translator template results in a delay in the response of the INVERTER independent of the value of $\rho$, while a decreased amplitude with increasing translator template is only observed for a value of $\rho$ close to 1. Hence, the overall dynamics of the heuristic model qualitatively agree with experimental data for a value of $\rho$ close to 1, indicating low inherent retroactivity arising from the load of the translator module. Thus, by constructing a minimal model we could identify properties significantly contributing to the dynamics of our system, and specifically determine and understand the origin of retroactivity.

**Connecting the translator module to a memories circuit.** Biochemical circuits with specific topology–function relationship inside cells are interconnected to downstream processes, and thereby regulate the time-dependent control of protein production. Analogous to the hierarchical layers of signal generation and processing in natural cells, we next explored the possibility to engineer and implement orthogonal control of two enzymatic actuators regulated by a synthetic bistable switch[25]. The two-state switchable network, as described in ref. [25], was constructed by joining two complementary INVERTER circuits (Fig. 4a) giving a symmetrical topology in which two autocatalytic modules dynamically repress each other. The core of the network consists of four templates, including the mutually exclusive autocatalytic templates $\alpha t o \alpha$ and $\beta t o \beta$ that produce key species $\boldsymbol{\alpha}$ and $\boldsymbol{\beta}$, and the inhibitory templates $\alpha t o i \beta$ and $\beta t o i \alpha$ from which inhibitors are produced cross-sequestering the autocatalytic templates. The network is defined to be in the α-state when the concentration of $\boldsymbol{\alpha}$ is high and $\boldsymbol{\beta}$ is low as a result of high activity of $\alpha t o \alpha$, which represses the autocatalytic $\beta t o \beta$ node via template $\alpha t o i \beta$. Likewise, the network is defined to be in the β-state when the concentration of $\boldsymbol{\beta}$ is high and $\boldsymbol{\alpha}$ is low corresponding to high activity of $\beta t o \beta$, which represses the autocatalytic $\alpha t o \alpha$ node via $\beta t o i \alpha$. Furthermore, two more templates $\gamma t o \alpha$ and $\delta t o \beta$ are included that serve as receivers for external inputs $\boldsymbol{\gamma}$ and $\boldsymbol{\delta}$ resulting in a long-lasting pulse of $\boldsymbol{\alpha}$ or $\boldsymbol{\beta}$, unbalancing the circuit and stimulating the network to switch (Supplementary Fig. 4). Orthogonal control of enzymatic actuators by the bistable switch can be achieved by coupling two distinct translator modules to primers $\boldsymbol{\alpha}$ and $\boldsymbol{\beta}$. To validate the activation of the translator modules by these species, a system was constructed in which first a single translator module, $\alpha t o \sigma$ or $\beta t o \sigma$, was coupled to $\boldsymbol{\alpha}$ or $\boldsymbol{\beta}$, respectively. Indeed, experimental results of the translator coupled to $\boldsymbol{\alpha}$ or $\boldsymbol{\beta}$ revealed that the production rate of translator output follows the dynamics of the switch (Supplementary Fig. 5). To assess the effect of

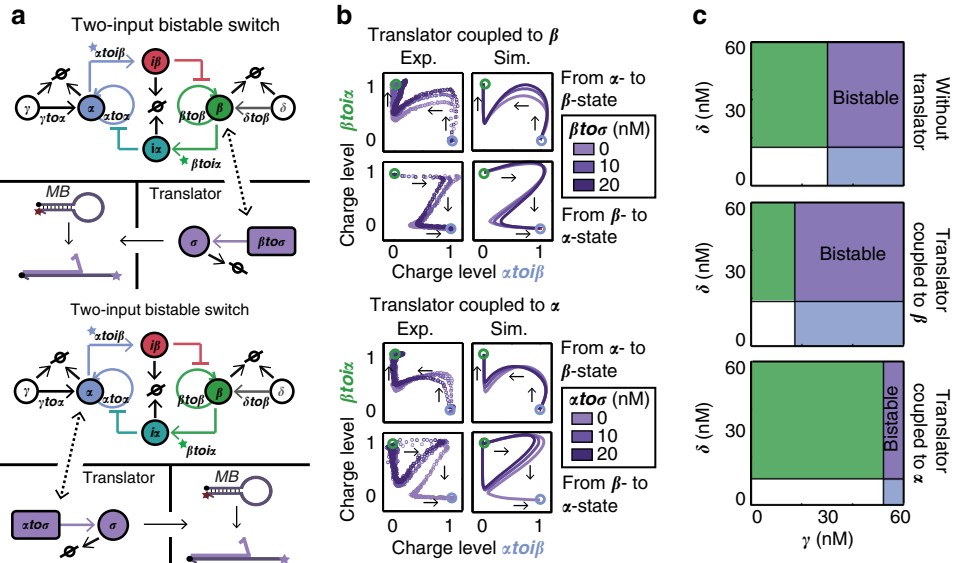

**Fig. 4** Characterizing retroactivity from coupling of the translator module to the memories circuit. **a** Schematic illustration of the system, in which the translator module is coupled to $\alpha$ or $\beta$ of the PEN-based bistable switch. The core of the bistable switch consists of four templates, including the autocatalytic templates $\alpha to\alpha$ and $\beta to\beta$ and the inhibitory templates $\alpha to\beta$ and $\beta to\alpha$. The network switches between states upon injection of $\gamma$ and $\delta$ which are received by templates $\gamma to\alpha$ and $\delta to\beta$. The dynamics of the bistable switch are followed via N-quenching using templates $\beta to\alpha$ and $\alpha to\beta$ which are 3'-end labeled with a DY530 and FAM fluorophore, respectively. **b** Experimental (Exp.) and simulated (Sim.) phase diagrams for a concentration range of translator template coupled to $\alpha$ or $\beta$. Experiments were carried out as described in the Methods using 20 nM $\beta to\alpha$, 15 nM $\alpha to\beta$, 24 nM $\beta to\beta$, 10 nM $\alpha to\alpha$, $\gamma to\alpha$, and $\delta to\beta$, 15 U mL$^{-1}$ Bst 2.0 WarmStart DNA polymerase, 10 U mL$^{-1}$ Nt. bstNBI, and 200 nM ttRecJ. The switch was either equilibrated to its $\alpha$-state and 30 nM $\delta$ was injected for switching to the $\beta$-state or the switch was equilibrated to its $\beta$-state and 30 nM $\gamma$ was injected for switching to the $\alpha$-state. The charge level is the normalized fluorescence of the signal of DY530 and FAM fluorophores, which is 0 in the absence of template's input primer and 1 at the steady-state value of primer $\beta$ and $\alpha$, respectively. The blue and green circles represent the $\alpha$-state and $\beta$-state, respectively. Simulations were performed using the heuristic model (Supplementary Notes) and the traces were converted to normalized units (n.u.) by normalizing $\alpha$ and $\beta$ to their steady-state concentrations. **c** Bifurcation diagrams of the switch in isolation and with 10 nM of translator module coupled to $\beta$ or $\alpha$ as a function of inputs $\gamma$ and $\delta$ obtained using the heuristic model (Supplementary Notes). The monostable domains of $\alpha$ and $\beta$ are shown in blue and green, respectively, while the bistable domain is shown in purple

retroactivity arising from the additional load of the translator module to the switch, we systematically increased the concentration of translator template coupled to either $\alpha$ or $\beta$ and switched the network from the $\alpha$-state to the $\beta$-state and conversely (Fig. 4b). The experimental trajectories show that we are able to switch the network both ways when the translator template is coupled to $\beta$. Furthermore, we were able to switch the network from the $\alpha$-state to the $\beta$-state when the translator is coupled to $\alpha$. However, the trajectories of switching the network from the $\beta$-state to the $\alpha$-state in this case show an initial increase in $\alpha$ after applying a pulse of $\gamma$, followed by a return to the $\beta$-state, indicating failure of switching to the $\alpha$-state. To obtain a fundamental understanding of these observations, the heuristic model that describes the INVERTER was adapted to the topology of the bistable switch (Supplementary Notes, Supplementary Figs. 18, 19, 24–27, 29). The trajectories obtained by the theoretical model correlated well with the experimental results (Fig. 4b and Supplementary Fig. 5). Interestingly, while $\rho$ was close to 1 for the INVERTER circuit, a value of 0.4 was obtained for the bistable switch, indicating an increased inherent retroactivity from the translator module compared to the INVERTER circuit likely due to a change in the system-dependent enzyme competition between polymerase and nickase (Supplementary Fig. 30). To analyze the effect of retroactivity, we computationally determined the bistable regime using the concentration of $\gamma$ and $\delta$ as bifurcation parameters in the absence of translator module and when the translator module is coupled to $\alpha$ or $\beta$ (Fig. 4c). The bifurcation diagram of the switch isolated from the translator module shows an asymmetry to the inputs as more $\gamma$ than $\delta$ is

required to obtain bistability, indicating a stronger preference of the $\beta$-state as also observed from the separatrix and computed switching planes (Supplementary Figs. 22–29). As previously noted, this imbalance can be explained by asymmetrical kinetics arising from differences in DNA hybridization Gibbs free energy (Supplementary Table 2)[25]. Coupling of the translator template to $\beta$ results in a shift in the separatrix and switching plane in favor of the $\alpha$-state (Supplementary Figs. 26a, 29). As a result, a decrease in asymmetry to the inputs $\gamma$ and $\delta$, and consequently an increase in bistable domain is observed indicating that, counterintuitively, coupling of a load to a bistable network can enhance the robustness of the upstream circuit by the retroactivity from the load. Contrary, the computed bifurcation diagram obtained by coupling of the translator module to $\alpha$ shows a decrease in the range of inputs that generate bistability caused by a shift in the separatrix in favor of the $\beta$-state (Supplementary Figs. 26a, 29) further increasing the asymmetry of the two states. Hence, retroactivity resulting from the additional load of the translator template to $\alpha$ narrows the parameter region, mostly by a shift in concentration of $\gamma$, for which bistable behavior is observed. Our theoretical model predicts that bistability can be recovered at high concentrations of input $\gamma$ (>50 nM). Indeed, experimental results show switching from the $\beta$-state to the $\alpha$-state with 10 nM of the translator module coupled to $\alpha$ upon injection of 50 nM $\gamma$ (Supplementary Fig. 6) in accordance with our theoretical predictions. In summary, retroactivity from coupling of the translator module to $\alpha$ or $\beta$ can either increase or decrease the input range for which bistable behavior can be observed, which depends on the asymmetry of the switch in isolation. Notably, the

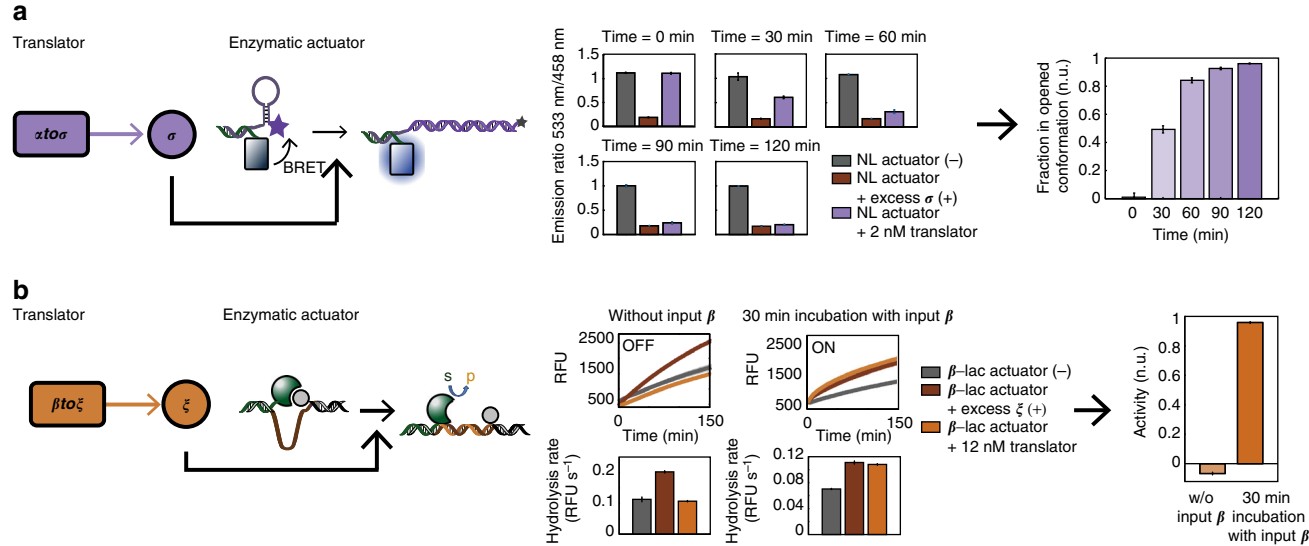

**Fig. 5** The control of enzymatic actuators by the translator module. **a** Schematic illustration (left) and experimental results (right) of controlling a NanoLuc (NL)-based actuator by the translator module. Experiments were performed using 2 nM of **αtoσ**, 5 nM of the NanoLuc-based actuator, 15 U mL⁻¹ Bst 2.0 WarmStart DNA polymerase, 10 U mL⁻¹ Nt. bstNBI, and initiated with 30 nM **α**. The opening of the stem-loop structure of the NanoLuc-based actuator was quantified at intervals of 30 min by measuring the BRET ratio between the NanoLuc donor (em. = 458 nm) and FAM acceptor dye (em. = 533 nm). The translator module was omitted for negative (−) and positive (+) controls and excess of DNA strand **σ** was added for the positive controls. Experiments were performed in triplicate and the fraction in opened conformation in normalized units (n.u.) was calculated by subtracting the mean BRET ratio of the positive controls and normalizing to the mean BRET ratio of the negative controls. Error bars and shaded areas represent the standard error of the mean of the experiments. **b** Schematic illustration (left) and experimental results (right) of the activation of the self-inhibitory TEM1 β-lactamase (β-lac) actuator by the translator module. Experiments were performed using 12 nM **βtoξ**, 2.5 nM TEM1 β-lactamase/BLIP actuator, 15 U mL⁻¹ Bst 2.0 WarmStart DNA polymerase, 10 U mL⁻¹ Nt. bstNBI, and initiated with 30 nM **β**. The activity of TEM1 β-lactamase was measured at time = 0 min prior to initiation with **β** and 30 min after activation of the translator module and was quantified by measuring the hydrolysis rate of fluorogenic substrate CCF2-FA obtained from the linear regime of the fluorescent time traces. The translator module was omitted for negative (−) and positive (+) controls and excess of DNA strand **ξ** was added for the positive controls. Experiments were performed in triplicate and the activity in normalized units (n.u.) was calculated by subtracting the mean hydrolysis rate of the negative controls and normalizing to the mean hydrolysis rate of the positive controls. Error bars and shaded area's represent the standard error of the mean of the experiments

retroactivity that arises from coupling of the translator module to **α** is relatively large compared to coupling to **β**, as visualized by the larger shift in the bistable domain. While the intrinsic retroactivity constant $\rho$ and the concentration of translator module was equal for both states of the switch, the dissociation rate constant of **α** is smaller than that of **β** arising from a lower equilibrium dissociation constant (Supplementary Table 2) accounting for the larger retroactivity. We validated this by computing the separatrices for different values of $K_{L\alpha}$ or $K_{L\beta}$ showing an increased shift with decreasing dissociation constant (Supplementary Fig. 26).

**Control of enzymatic actuators by the translator module.** Having established the translator module as a versatile method to translate short ssDNA from the upstream circuit to long DNA strands with minimal retroactivity, we next investigate the possibility to control enzymatic actuators by the translator module. To this end a bioluminescent actuator and a self-inhibitory TEM1 β-lactamase construct were used (Fig. 5). The bioluminescent actuator is based on a previously reported design[31] and consists of a NanoLuc enzyme conjugated to an oligonucleotide, which hybridizes to a template 3′-end labeled with a FAM fluorophore (Fig. 5a). In the closed state this template forms a stem-loop structure, which brings the FAM fluorophore and NanoLuc in close proximity resulting in bioluminescence resonance energy transfer (BRET) between the NanoLuc donor and FAM acceptor dye. Transient opening of the stem-loop structure of the NanoLuc-based actuator was accomplished by the translator template **αtoσ** producing activator

strand **σ**, which hybridizes to the loop of the NanoLuc-based actuator, and thereby via strand displacement disrupts the stem structure. In accordance, the experimental data show a gradual decrease in BRET ratio after initiation of the translator module, which can be followed in time. Positive (+) and negative (−) controls were run in parallel to account for the decrease in BRET efficiency over time (Methods). The rate of opening of the stem-loop structure of the enzymatic actuator can be fine-tuned by the concentration of translator module, which scales with the production rate of its output strand (vide supra). Likewise, the activity of the TEM1 β-lactamase enzyme was controlled by an orthogonal translator module (Fig. 5b). β-lactamases are enzymes produced by bacteria to provide antibiotic resistance and are often used as reporter enzymes or to install antibiotic resistance[38]. Using a previously reported design[9], the activity of TEM1 β-lactamase is controlled by modulation of the interaction of this enzyme with the β-lactamase inhibitor protein BLIP[9]. Specifically, the proteins are conjugated to different oligonucleotides that hybridize to a template connecting the enzyme and inhibitor. Activation of TEM1 β-lactamase is achieved by translator template **βtoξ** producing activator strand **ξ**, which hybridizes to the loop of the self-inhibitory TEM1 β-lactamase actuator, and thereby separating enzyme and inhibitor. The activity of TEM1 β-lactamase was determined by measuring the hydrolysis rate of a fluorescent substrate. Positive (+) and negative controls (−) were run in parallel to account for the loss in activity of TEM1 β-lactamase in the PEN-toolbox buffer (Methods). As observed from the experimental results, the activity of the TEM1 β-lactamase was equal to the background activity prior to initiation of

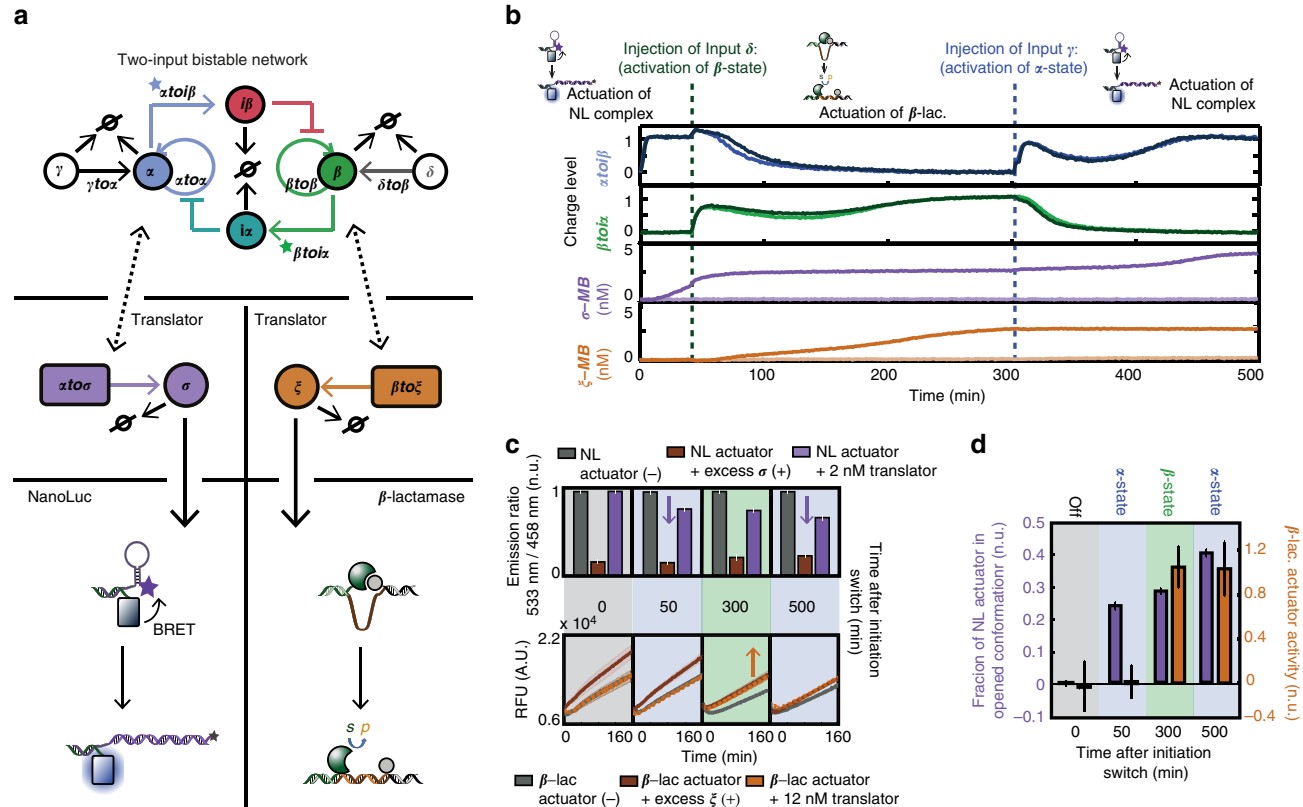

**Fig. 6** Control of two orthogonal enzymatic actuators by a switchable memories circuit. **a** Schematics of the experiment in which the switch controls a NanoLuc-based actuator and a self-inhibitory TEM1 $\beta$-lactamase construct. **b–d** Results of the experiments, carried out as described in the Methods, using 20 nM $\boldsymbol{\beta toi\alpha}$, 15 nM $\boldsymbol{\alpha toi\beta}$, 24 nM $\boldsymbol{\beta to\beta}$, 10 nM $\boldsymbol{\alpha to\alpha}$, $\boldsymbol{\gamma to\alpha}$, and $\boldsymbol{\delta to\beta}$, 15 U mL$^{-1}$ Bst 2.0 WarmStart DNA polymerase, 10 U mL$^{-1}$ Nt. bstNBI, and 200 nM ttRecJ. Reactions were performed in presence of the actuators or MBs. The switch was initiated with 1 nM $\boldsymbol{\alpha}$. **b** The graphs show the dynamics of the switch and the production of $\sigma$ and $\xi$ measured using MBs (5 nM $\boldsymbol{MB_\sigma}$ and 2.5 nM $\boldsymbol{MB_\xi}$) (Supplementary Fig. 17) in absence (light color) and in presence (dark color) of the translator modules (2 nM $\boldsymbol{\alpha to\sigma}$ and 12 nM $\boldsymbol{\beta to\xi}$). The charge level is the normalized fluorescence of the signal of DY530 and FAM fluorophores, which is 0 in the absence of template's input primer and 1 at the steady-state value of $\beta$ and $\alpha$, respectively. The dotted lines show the time points at which 30 nM of the Inputs $\boldsymbol{\delta}$ and $\boldsymbol{\gamma}$ were added. In parallel, experiments were run where the MBs were replaced with the enzymatic actuators (5 nM of the NanoLuc-based actuator and 2.5 nM TEM1 $\beta$-lactamase actuator). **c, d** The state of the actuators was measured at four time points including negative (−) and positive (+) controls (Supplementary Figs. 13, 14 and Methods). Error bars and shaded area's represent the standard error of the mean of the experiments. Experiments were performed in plurality (> 3) and at three different days. **c** The bar graphs displaying the BRET ratio were normalized to the mean of the negative controls for a clear visualization (Supplementary Fig. 14 displays the raw data). **d** The activity or fraction in opened conformation of the actuators were calculated by normalizing to positive and negative controls (Methods)

the translator module, while almost complete activation was achieved after 30 min of incubation with initiator $\beta$. Because of the high binding affinity of the activator strands $\sigma$ and $\xi$, the opening of the stem-loop structure of the NanoLuc-based actuator and the activation of the self-inhibitory TEM1 $\beta$-lactamase construct are irreversible. These results show that the translator module is compatible with the NanoLuc-based actuator and the self-inhibitory TEM1 $\beta$-lactamase construct, and, importantly, reveals it to be a flexible method for the controlled and efficient activation of these actuators.

**Orthogonal control of enzymatic actuators by switch circuit.** Having characterized the translator module as a generic method to control enzymatic actuators and to translate short oligomers to long output strands with minimal retroactivity, we implemented the control of the enzymatic actuators by the PEN-based bistable switch. First, we showed the control of either the self-inhibitory TEM1 $\beta$-lactamase construct (Supplementary Fig. 9) or the NanoLuc-based actuator (Supplementary Figs. 10, 11) by one of the two states of the switch. Thereafter, we investigated whether it is possible to implement orthogonal control of the enzymatic actuators by the two states of the switch. To this end, we

constructed a system in which the NanoLuc-based actuator was controlled by coupling of $\alpha$ to translator template $\boldsymbol{\alpha to\sigma}$, while the self-inhibitory TEM1 $\beta$-lactamase construct was controlled by coupling of $\beta$ via translator template $\boldsymbol{\beta to\xi}$ (Fig. 6a). Because of technical reasons the control of the actuators could not be directly followed in the samples (Methods). Therefore, the activity of TEM1 $\beta$-lactamase and the conformation of the NanoLuc-based actuator were determined at four different states of the switch by taking aliquots and measuring the hydrolysis rate or BRET ratio immediately after addition of the NanoLuc and $\beta$-lactamase substrates. We measured the production rate of $\sigma$ and $\xi$ using parallel experiments in which the enzymatic actuators were replaced with two orthogonal MBs. Control experiments were performed showing no interference of the MBs and actuators with the bistable switch (Supplementary Figs. 15, 16). Furthermore, experiments were performed in the absence of the orthogonal translator modules to quantify the retroactivity to the dynamics of the bistable switch that arises due to the additional load. Indeed, the results displayed in Fig. 6b (and Supplementary Fig. 12) reveal that the retroactivity that arises from coupling of the orthogonal set of translator modules to the dynamics of the upstream bistable circuit is low. Importantly, the results in Fig. 6b

show that production of $\sigma$ is initiated after activation of the $\alpha$-state of the switch, while the production rate of $\xi$ is zero indicating that the switch has adopted the $\alpha$-state. In agreement, we observe a decrease in BRET ratio 50 min after initiation of the switch as shown in the bar graphs in Fig. 6c. In contrast the TEM1 $\beta$-lactamase-based actuator is not activated in the $\alpha$-state as shown by the overlapping fluorescent traces of the hydrolysis of CCF2-FA of the samples and negative controls (Fig. 6c). For a clear visualization the BRET ratio and hydrolysis rate of the samples were normalized to positive and negative controls (Methods). Figure 6d shows an increased opening of the stem-loop structure of the NanoLuc-based actuator toward ~1/4 of its fully opened conformation 50 min after initiation of the switch. Next, we injected input $\delta$ to switch the network to the $\beta$-state, as observed from the biphasic evolution of the charge levels of $\alpha to i\beta$ and $\beta to i\alpha$. This results in the downstream activation of $\beta to \xi$ and the production of $\xi$, while the production of $\sigma$ ceases. The hydrolysis rate of the TEM1 $\beta$-lactamase construct and BRET ratio of the NanoLuc-based actuator including (−) and (+) controls were measured again 250 min after injection of $\delta$. The results show that the TEM1 $\beta$-lactamase enzyme is completely activated, while the stem-loop structure of the NanoLuc-based actuator has only slightly opened. Switching the network back to the $\alpha$-state by injection of input $\gamma$ results in continued production of $\sigma$, while production of $\xi$ ceases. Likewise, an increase in opened conformation of the NanoLuc-based actuator is observed 200 min after injection of $\gamma$, while the TEM1 $\beta$-lactamase enzyme stays at its completely activated state. These results demonstrate that we are able to successfully time and control the activity of enzymatic actuators by the dynamics of the bistable switch by judicious design of orthogonal translator modules with low retroactivity.

## Discussion

Our work shows the possibility of connecting enzymatically enriched DNA circuits that are capable of displaying higher-order regulatory behavior to a variety of biochemical actuators in vitro, such as a TEM1 $\beta$-lactamase and a luciferase-based system. Previously, non-enzymatic, nucleotide-based logic circuits have been used to engineer autonomous cell-free systems capable of programmable manipulation of protein activity in vitro[15, 39, 40]. While non-enzymatic circuits are capable of basic information processing functions such as logic operations, amplification, and input thresholding, enzymatically driven systems can display a much broader range of system-level behaviors such as bistability, oscillations, and perfect adaptation[41, 42]. Each of these dynamic regulatory behaviors comes with a unique set of information processing functions. For example, bistable circuits in the living cell can generate sharp input thresholds and can either reversibly or irreversibly switch to an activated state[43]. In addition, bistable gene regulatory networks can also act as dynamic noise filters by ignoring transient changes in the input signal[44]. Perfect adaptation, another type of non-equilibrium dynamic behavior, is an important feature of cellular regulation and is typically used to generate homeostatic behavior[45]. Finally, oscillatory circuits in living cells are not only used for time-keeping functions but can also transmit information via coding and decoding of temporal signaling patterns[46]. These examples indicate that protein activity controlled via enzymatically enriched nucleic acid-based computing systems can yield autonomous cell-free systems with more advanced information processing functions than is currently possible.

While modularity has often been cited as a key advantage of nucleic acid-based chemical systems, our work reveals that in order to reliably connect an upstream DNA-based network to a downstream enzymatic load, retroactivity has to be taken into account. Our theoretical analysis shows that biochemical loads can bias the dynamical properties of bistable switches based on reciprocal inhibition in a manner that depends critically on the strength of the two states in the absence of load. In the living cell, bistability is found in many important gene regulatory networks and signal transduction pathways that regulate cell proliferation[47], cell-fate determination[48], and Ras activation[49]. However, in many cases the mathematical models that describe these regulatory circuits do not incorporate the effects of downstream components while these are certainly present. In an insightful study, Prasad and co-workers[50] theoretically analyzed the effect of downstream loads on bistable genetic and signaling switches and found that the addition of load changes the underlying potential energy landscape skewing it in favor of the unloaded side. In addition, the authors found that in some cases the additional downstream load can abrogate bistable dynamics. Our experimental results on the effect of downstream loads on the DNA-based bistable switch indeed confirm these predictions, as we observe failure of switching dynamics when the load is coupled to the weaker $\alpha$-state of the network. Furthermore, because the PEN-based bistable switch is inherently asymmetric due to a difference in binding affinity of the $\alpha$ and $\beta$ primers to their corresponding complementary sequences, the effect of a downstream load to each of the two states is also different. In the absence of load, the $\beta$-state is stronger than the $\alpha$-state meaning the concentration of input $\delta$ needed to switch the network to the $\beta$-state is less compared to the concentration of input $\gamma$ that is needed to switch the system to the $\alpha$-state. The theoretical analysis shows that coupling of a downstream load to the weaker $\alpha$-state results in further weakening of the $\alpha$-state and a concomitant narrowing of the concentration range of $\delta$ and $\gamma$ for which bistability can observed. However, when the load is applied to the stronger $\beta$-state of the switch, the potential energy landscape becomes more symmetrical resulting in a larger input parameter range for which bistable behavior can be observed. While in general low retroactivity is desired, our work shows that retroactivity not necessarily has a negative effect.

In summary, we have shown how a cell-free bistable switch can be used to time and control protein-based activity by engineering a new module enabling connection of the upstream circuit and downstream actuators, taking into account proper design constraints. By allowing the orthogonal integration of distinct molecular platforms our work represents a key step for the development of cell-free biochemical systems of increasing chemical complexity, providing the potential for new insights in cellular networks and ultimately the construction of synthetic cells.

## Methods

**Materials.** Oligonucleotides (Supplementary Table 1) were obtained from Integrated DNA Technologies (IDTDNA) or Biomers and were purified using high-performance liquid chromatography. Templates which are not 5′-end labeled with a fluorophore or a quencher have three phosphorothioate backbone modifications at the 5′-end preventing them from degradation. Furthermore, since 3′-OH can be extended by DNA polymerase, the templates were ordered with a phosphate modification at their 3′-end to prevent circuit leakage. Templates at which primers are produced inevitably have an additional nickase recognition site at the template's output site. Based on previous work[25], to decrease the affinity of the nicking enzyme for the output site, the thymine base in the nickase recognition site at the template's output was replaced with a uracil base. Concentrations of DNA were verified using UV-spectrophotometry. The nicking enzyme and polymerase were obtained from New England Biolabs (NEB), while ttRecJ, a thermophilic equivalent of the RecJ enzyme from *Thermus thermophilus*, was obtained from Dr. A. Estévez-Torres.

**PEN-based experiments.** Throughout the study, reactions of a total volume of 20 μL were assembled in a master mix, containing 20 mM Tris-HCl, 10 mM KCl, 50 mM NaCl, 10 mM $(NH_4)_2SO_4$, 8 mM $MgSO_4$, 0.1% Triton ×-100, 400 μM of each deoxyribonucleotide triphosphates (NEB), 0.1% Synperonic F108 (Sigma

Aldrich), 2 µM Netropsin (Sigma Aldrich), 1 mg/mL bovine serum albumine (BSA; NEB), 4 mM Dithiothreitol (DTT), and a pH of 8.8. A 4× stock solution of the master mix was prepared, excluding BSA and DTT, which were added during reaction assembly together with enzymes, oligonucleotides, and enzymatic actuators (which were pre-assembled as described vide infra). The nicking enzyme Nt. BstNBI and Bst. 2.0 WarmStart DNA polymerase were obtained from NEB, while ttRecJ, a thermophilic equivalent of the RecJ enzyme from *Thermus thermophilus*, was obtained from André Estévez-Torres. The activity of each batch of ttRecJ was determined using the experiments as described in Supplementary Fig. 20 and batch-to-batch variations were compensated by changing the enzyme concentration. For experiments in which injections were done during the experiment an oil layer (15 µL) was used to prevent a shift of the signal after injection. Experiments were performed at a temperature of 42 °C and fluorescence was recorded over time (CFX96 PCR machine).

Experimental data of FAM and DY530 were handled by subtracting the raw data by a baseline curve. This baseline curve was measured for both the FAM and DY530 channels in presence of the inhibition templates and in absence of amplification of $\alpha$ and $\beta$, respectively. Thereafter, the signal of DY530 and FAM fluorophores were normalized to the charge levels of $\beta to i \alpha$ and $\alpha to i \beta$ which is 0 in the absence of template's input primer and 1 at the steady-state value of primer $\beta$ and $\alpha$, respectively. The fluorescence of Cy5 and ROX fluorophore attached to the MBs were converted to concentration of DNA strand $\sigma$ and DNA strand $\xi$, respectively, using a standard curve.

**Assembling of the NanoLuc-based actuator**. Protein expression, conjugation, and purification of NanoLuc was carried out as reported[31]. In short, a cysteine was genetically inserted in the C-terminus of NanoLuc via site-directed mutagenesis and the plasmid was transformed into *E. coli* BL21(DE3). Subsequently, the cells were cultured in LB medium and protein expression was induced at $OD_{600} = 0.6$ by the addition of 100 µM IPTG. After overnight expression at 18 °C the cells were lysed by centrifuging the cells at $10,000 \times g$ for 10 min and subsequently dissolving the pelleted cells in BugBuster protein extraction reagent (Novagen) and Benzonase endonuclease (Novagen). The lysed cells were subsequently centrifuged ($40,000 \times g$ for 40 min) to obtain the soluble fraction, from which NanoLuc was purified using $Ni^{2+}$-affinity chromatography.

Amine-modified oligonucleotide ($ODN_{NL}$) was dissolved in PBS (100 mM NaPi, 150 mM NaCl, pH 7.2) to a final concentration of 1 mM and mixed with 20 equivalents of Sulfo-SMCC (freshly dissolved in DMSO to 20 mM) and incubated for 2 h at room temperature while shaking at 850 rpm. Subsequently, the excess Sulfo-SMCC was removed by extracting the maleimide-activated oligonucleotide by three rounds of ethanol precipitations and the oligonucleotide was dried under vacuum. Prior to oligonucleotide conjugation, NanoLuc was buffer exchanged to 100 mM sodium phosphate, pH 7.0 by gel-filtration (PD-10 desalting column) and directly added to a 3-fold molar excess of maleimide-activated oligonucleotide and allowed to react for 2 h at room temperature while shaking at 850 rpm. Subsequently, the oligonucleotide-NanoLuc conjugate ($NL$-$ODN_{NL}$) was purified by consecutive $Ni^{2+}$-affinity chromatography to remove excess oligonucleotide and anion-exchange chromatography to remove unconjugated protein. The NanoLuc actuator was hybridized prior to use by mixing together 100 nM $NL$-$ODN_{NL}$ and 120 nM $NLlink$-$\sigma$ and left at room temperature for at least 1 h.

**Assembling of the β-lactamase actuator**. Protein expression, conjugation, and purification of β-LactamaseE104D and BLIP were carried out as previously reported[9]. Furthermore, the complex was hybridized prior to use by mixing together 100 nM βlac-ODN, 200 nM BLIP-ODN, and 120 nM βlacLink-ε and left at room temperature for at least 1 h.

**Measuring the conformational state of the NanoLuc-based actuator**. Because of technical reasons the hydrolysis rate of the TEM1 β-lactamase actuator and BRET ratio of the NanoLuc-based actuator could not be measured over time in the samples directly. NanoGlo (substrate for NanoLuc) is not compatible with the PEN toolbox and somehow has an effect on the dynamics of the PEN-based network. Moreover, BRET detection could not be performed by the CFX96 PCR machine in which the switching behavior was followed. Even so, the excitation and emission filters of the CFX96 PCR were not compatible with the excitation and emission wavelength of converted CCF2-FA substrate of TEM1 β-lactamase. Hence, the conformation of the NanoLuc-based actuator, controlled by the translator module in absence (Fig. 5) or presence (Fig. 6, Supplementary Fig. 10) of the switch, was determined by taking 16 µL of the sample (out of 20 µL) and transferred to a 396-well plates prefilled with 20 µL of master mix and 25 µL oil to prevent condensation. Negative and positive controls were carried out in parallel. Negative controls were run to quantify the decrease in BRET efficiency of the NanoLuc-based actuator at different time intervals, and therefore the translator module(s) were omitted in the reactions of the negative controls (all other components were exactly the same as in the samples). Positive controls were run to quantify the BRET signal for maximal opening of the NanoLuc-based actuator at different time intervals. For positive controls the translator module(s) were also omitted in the reactions and excess of DNA strand $\sigma$ (200 nM) was added to the wells plate (all other components were exactly the same as in the samples). Thereafter,

the plate was put in the centrifuge at 1000 rpm for 30 s. After 10 min incubation at 42 °C in the plate reader (Tecan, Spark 10 M) 4 µL of 50× diluted Nano-Glo (Promega) was added after which the mixtures were mixed and an emission spectrum (from 400 to 650 nm) was measured. The BRET ratio between 533 and 458 nm was calculated and the fraction of opened stem-loop structure of the NanoLuc-based actuator was calculated by subtracting the mean BRET ratio of the positive controls and normalizing to the mean BRET ratio of the negative controls.

**Measuring the activity of the β-lactamase actuator**. Because of technical reasons the hydrolysis rate of the TEM1 β-lactamase actuator and BRET ratio of the NanoLuc-based actuator could not be measured over time in the samples directly. NanoGlo (substrate for NanoLuc) is not compatible with the PEN toolbox and somehow has an effect on the dynamics of the PEN-based network. Moreover, BRET detection could not be performed by the CFX96 PCR machine in which the switching behavior was followed. Even so, the excitation and emission filters of the CFX96 PCR were not compatible with the excitation and emission wavelength of converted CCF2-FA substrate of TEM1 β-lactamase. Hence, the activity of TEM1 β-lactamase, activated by the translator module in absence (Fig. 5) or presence (Fig. 6, Supplementary Fig. 9) of the switch, was determined by taking 16 µL of the sample (out of 20 µL) and transferred to a 396-well plates prefilled with 20 µL of master mix and 25 µL oil to prevent condensation. Negative and positive controls were carried out in parallel. Negative controls were run to account for the change in activity of unactivated TEM1 β-lactamase/BLIP/$\xi$ complex in the PEN-toolbox buffer. For negative controls the translator module(s) were omitted in the reactions (all other components were exactly the same as in the samples). Positive controls were run to quantify the hydrolysis rate of maximal activated TEM1 β-lactamase/BLIP/$\xi$ complex (Supplementary Figs. 7, 8). For positive controls the translator module(s) were also omitted in the reactions and excess of DNA strand $\xi$ (100 nM) was added to the wells plate (all other components were exactly the same as in the samples). Thereafter, the plate was put in the centrifuge at 1000 rpm for 30 s. After 10 min incubation at 42 °C in the plate reader (Tecan, Safire) 4 µL of 20 µM CCF2-FA (Invitrogen) was added after which the mixtures were mixed and fluorescence (ex: 410 nm; em: 447 nm) was measured for at least 150 min. The hydrolysis rate of β-lactamase was determined by fitting the slope between 50 and 150 min. The activity was normalized by subtracting the mean hydrolysis rate of the negative controls and normalizing to the mean hydrolysis rate of the positive controls.

**Data availability**. The data sets generated during and/or analyzed during the current study are available from the corresponding author on reasonable request.

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

## Acknowledgements

This work was supported by European Research Council (ERC) Starting Grants (677313 BioCircuit and 280255 SwitchProteinSwitch), an ECHO-STIP grant from the Netherlands Organization for Scientific Research (NWO, 717.013.001), and funding from the Ministry of Education, Culture and Science (Gravity program, 024.001.035). The ttRecJ was a kind gift from Dr. A. Estévez-Torres.

## Author contributions

L.H.H.M. and A.J. performed experiments; L.H.H.M., E.S., and R.A.v.S. constructed the theoretical model; L.H.H.M. and E.S. analyzed the data. W.E. and M.M. developed the enzymatic actuators; L.H.H.M. and T.F.A.d.G. wrote the manuscript; T.F.A.d.G. supervised the research. All authors discussed the results and implications and commented on the manuscript at all stages.

## Additional information

**Competing interests:** The authors declare no competing financial interests.

