## [Peer Review File · Nature Communications]

Reviewers' Comments:

Reviewer #1:

Remarks to the Author:

This is a revised version of a manuscript which I previously reviewed. In this paper, the authors integrate a bistable switch powered by the PEN DNA toolbox (a framework to program enzymatic DNA reaction networks) with downstream enzymatic actuation. The paper is impressively thorough, as the authors have meticulously modelled, tested and optimized each component of the system. This paper also expands the breadth of DNA circuits, because most reported DNA circuits tend to work in autarky (taking DNA strands as input and churning out other strands as output, or alternatively fluorescence signals). As such I hope that it will encourage others to work on this area so as to widen the applicability of DNA circuits.

I made 28 major or minor comments during the first round of review. I applaud the diligence of the authors, who addressed all of them carefully. They performed the crucial additional experiments that I suggested, showing that the observed degradation of performance is caused by the PEN DNA toolbox buffer, rather than by some design problems. The authors also give now more emphasis to raw data, which will help the reader to form an educated opinion. They have satisfactorily reworked the text and figure to reflect the replies they made to my comments. As for as my review is concerned, I do not have anything to add and warmly recommend acceptance of this paper.

I have also read comments voiced by the other reviewers. I find them harsh. Reviewer 1 averred that the integration of synthetic DNA circuits with enzymatic actuators is trivial. Had it been so, the whole paper would not have totalled 50 pages of SI. Integration is not just a matter of blindly mixing DNA strands and enzymes, and hoping that somehow, something will work. It is a question of finding the right balance between all components to achieve optimal function. For instance, the secondary effects of downstream modules on upstream ones must be carefully balanced, or insulated- which requires not only careful mathematical modelling, but also the design of new components.

Reviewer 2 gratuitously contends that the "kinetics of all DNA devices are known". Well, the kinetics of DNA devices is often known a posteriori, once a phenomenological model with appropriate fudge factors has been fit to experimental time traces. But predicting from scratch the kinetics of an arbitrary DNA system is a daunting task, and that is exactly why we still do experiments. If the kinetics of all DNA devices was known, the field of DNA computing would have come to a halt circa 1995-2000. At this time, the field enjoyed a huge boom in "recipe" papers that proposed a myriad of applications, provided the system worked exactly like its kinetic model predicted. Needless to say that it took almost a decade to get practical DNA circuits, for example with the seminal work of Benenson, Stojanovic, Winfree, and others, who not only simulated DNA system, but also built them